# The Association of Dietary Polyamines with Mortality and the Risk of Cardiovascular Disease: A Prospective Study in UK Biobank

**DOI:** 10.3390/nu16244335

**Published:** 2024-12-16

**Authors:** Su Han, Mingxia Qian, Na Zhang, Rui Zhang, Min Liu, Jiangbo Wang, Furong Li, Liqiang Zheng, Zhaoqing Sun

**Affiliations:** 1Department of Cardiology, Shengjing Hospital of China Medical University, Shenyang 110004, China; hansu8866@126.com (S.H.); dafer1999@163.com (J.W.); 2School of Public Health, Shanghai Jiao Tong University School of Medicine, Shanghai 200025, China; qianmx2000@126.com (M.Q.); nazhang2021@shsmu.edu.cn (N.Z.); 3College of Public Health, Shanghai University of Medicine and Health Sciences, Shanghai 201318, China; gina_zr@126.com; 4Department of Epidemiology, School of Public Health, China Medical University, Shenyang 110122, China; 17725165216@163.com; 5School of Public Health and Emergency Management, Southern University of Science and Technology, Shenzhen 518055, China; lifr@sustech.edu.cn; 6Ministry of Education-Shanghai Key Laboratory of Children’s Environmental Health, Xinhua Hospital Affiliated to Shanghai Jiao Tong University School of Medicine, Shanghai 200092, China

**Keywords:** spermidine, spermine, putrescine, cohort study, coronary heart disease, stroke

## Abstract

Background: Polyamines, including spermidine (SPD), spermine (SPM) and putrescine (PUT), are essential for cellular physiology and various cellular processes. This study aimed to examine the associations of dietary polyamines intake and all-cause mortality and incident cardiovascular disease (CVD). Methods: This prospective cohort study included 184,732 participants without CVD at baseline from the UK Biobank who had completed at least one dietary questionnaire. Diet was assessed using Oxford WebQ, a web-based 24 h recall questionnaire, with polyamines intakes estimated from previous studies. Cox proportional models with restricted cubic splines were employed to investigate nonlinear associations. The primary endpoint was all-cause mortality or incident CVD (including CVD death, coronary heart disease and stroke). Results: During a median follow-up period of 11.5 years, 7348 (3.9%) participants died and 12,316 (6.5%) developed incident CVD. Polyamines intake showed nonlinear associations with all-cause mortality and incident CVD (P for nonlinear < 0.01). Compared to the lowest quintile group of dietary polyamines intake (≤17.4 mg/day), the quintile 2 to 5 groups demonstrated a reduced risk of all-cause mortality, with the lowest risk in quintile 2 group (>17.4–22.3 mg/day) (HR:0.82, 95% CI: 0.76–0.88). Similar results were observed for incident CVD, with the lowest risk in the quintile 4 group (>27.1–33.5 mg/day) (HR: 0.86, 95% CI: 0.82–0.92). Conclusions: We found that dietary polyamines intake was associated with a lower risk of all-cause mortality or incident CVD. Furthermore, our study identified an optimal range of dietary polyamines intake.

## 1. Introduction

Cardiovascular disease (CVD) is a leading cause of mortality, accounting for over 3.8 million deaths each year or 45% of all deaths across European society of cardiology member countries. Meanwhile, the disability resulting from ischemic heart disease or stroke makes up 22–24% of the total disability-adjusted life years [1]. Despite the advancement in therapeutic development, the socioeconomic and human burdens continue to rise with an aging population. A compelling body of evidence suggests that the composition of diet is closely related to disease onset and progression. In a prospective cohort study, greater dietary cholesterol and egg consumption were associated with an increased risk of overall and CVD mortality [2]. Furthermore, dietary total fat, compared to total carbohydrate, was inversely associated with total mortality [3]. Heightened attentions to dietary habits may offer a novel strategy to combat CVD.

Polyamines, including spermidine (SPD), spermine (SPM) and putrescine (PUT), are vital small polycationic metabolites crucial for various cellular processes such as chromatin structure modulation, transcriptional and translational regulation from oxidative damage, nucleic acid depurination and the regulation of multiple ion channels necessary for cell-to-cell communication [4,5]. Conversion among polyamines is possible. Ornithine decarboxylase (ODC), encoded by the ODC1 gene, is the rate-limiting step in polyamine biosynthesis and produces PUT from ornithine. Subsequently, PUT is converted to SPD and then SPM by two specific aminopropyltransferases, SPD synthase and SPM synthase, respectively [6]. Although gut bacteria contribute significantly to polyamine levels, dietary intake also plays an essential role in metabolism processes [7]. Dietary polyamines are present in a number of foodstuffs, including mushrooms, soybeans, natto, aged cheese and wheat germ. SPD, in particular, is linked to cardiac autophagy and mitophagy activation, lower subclinical plasma levels of the pro-inflammatory cytokine TNF/TNF-a, the enhancement of cardiac mitochondrial volume and respiration and the improvement of titin phosphorylation, which is a known molecular event that promotes the intrinsic elasticity of cardiomyocytes [8,9,10]. The present study looked at evidence on the oral supplementation of natural SPD (drinking water 3 mmol/L) and found that higher SPD intake was associated with extending the lifespan of C57BL/6J wild-type mice [11]. Furthermore, replenishing polyamine levels through dietary supplements in hypertensive Dahl salt-sensitive rats has been recently shown to postpone the onset of hypertensive heart disease and reduce arterial blood pressure [12]. Given the increasing mechanistic evidence of polyamines’ benefits, researchers proposed the assumption that dietary polyamines may help to reduce the risk of all-cause mortality. In the Bruneck study of 829 participants aged 45–84 y, higher SPD intake was linked to lower mortality during a follow-up of 20 years [13]. Comparable findings were found in the U.S. National Health and Nutrition Examination Survey (NHANES) study of 23,894 participants (mean aged 49 y, excluding participants with missing information on dietary SPD) [14]. However, researchers found contradictory conclusions in a cohort of 29,079 participants without coronary heart disease (CHD), stroke and cancer at baseline [15].

Based on the differing conclusions of previous studies on polyamines and all-cause mortality, and a lack of research on dietary polyamines and CVD risk, we relied on a population of approximately 500,000 participants in the UK Biobank, and up to five dietary questionnaires, to evaluate the relationship between dietary polyamines (SPD, SPM and PUT) and all-cause mortality or incident CVD. Additionally, we examined the potential influence of other established risk factors for CVD on the results.

## 2. Methods

### 2.1. Patients and Study Design

The UK Biobank serves as a vital health resource in the UK, with the objective of enhancing the prevention, detection and treatment of various diseases while promoting community well-being. Between 2006 and 2010, about 500,000 participants aged 37–73 were recruited into the UK Biobank [16,17].

Our study utilized data from 502,490 participants. Participants were excluded if they had CVD at baseline (n = 25,790), a lack of dietary data (n = 275,746), showed an anomalous energy range (defined as >20,934 KJ/day or <3349 KJ/day for male and >16,747 KJ/day or <2093 KJ/day for female) (n = 10,202) and displayed extreme values (beyond ±4 standard deviations (SD) from the mean) in dietary SPD, SPM and PUT data (n = 3320) [18]. Eventually, 187,432 participants were included in the study (Appendix A).

### 2.2. Exposure Assessment

Dietary consumption was evaluated through a self-reported, online questionnaire that included approximately 240 typical food and beverage items consumed over the previous 24 h. The 24 h dietary recall tool was initially introduced at the conclusion of the recruitment phase (2009–2010), and all the participants with available email addresses were invited to fill out the online survey four times during the years 2011 to 2012. The online survey demonstrated comparable outcomes to interviewer-led 24 h recalls in recording food and beverage consumption, and in predicting energy and nutrient levels, with an average Spearman correlation coefficient of 0.62 (ranging from 0.54 to 0.69) for macronutrients [19]. Each participant provided portion size details of their dietary intake. A nutrient database for dietary SPM, SPD and PUT was compiled from published data in previous publications [13,20,21,22,23,24,25,26,27,28,29]. Polyamines are defined as the sum of SPD, SPM and PUT based on their highest content and biological activity at present [30]. Each participant completed between one and five dietary questionnaires, allowing us to calculate their average daily polyamines intake. Dietary polyamines were categorized into quintiles: Q1: ≤17.4 mg/day; Q2: >17.4–22.3 mg/day; Q3: 22.3–27.1 mg/day; Q4: >27.1–33.5 mg/day; Q5: >33.5–98.9 mg/day.

### 2.3. Definitions of Covariates

A baseline touch screen survey was employed to evaluate several potential confounders: sex, age, ethnicity, education (categorized as university/college degree and other), smoking status (never/former/current), alcohol status (never/former/current), sleep duration (<7 h/day, 7–8 h/day and >8 h/day) and energy. The Townsend deprivation index (TDI), derived from postal codes and census data, served as a proxy for socioeconomic status, with higher scores indicating a greater socioeconomic disadvantage (classified as quintiles 1 being the least deprived, quintiles 2 to 4 as moderately deprived, and quintiles 5 as the most deprived) [31]. The body mass index (BMI) of each participant was calculated using weight and height measurements and categorized based on WHO guidelines: underweight (<18.5), normal weight (18.5 to <25), overweight (25 to <30), or obese (≥30). Participants were also classified based on their physical activity levels (high: ≥1200 (Metabolic Equivalent Task (MET) minutes per week for all activity), low: <1200 MET-min/week) according to global health recommendations [32]. Self-reported doctor diagnoses were used to determine comorbidities and medical history, which were confirmed during the in-person interview.

### 2.4. Polygenic Risk Score

The UK Biobank included the genome-wide genotype data (~805,000 markers) collected on all individuals in the cohort and its quality control procedures [33]. The genetic risk score for CHD and stroke were computed using genetic variants reported in the previous studies and involved 63 and 32 single nucleotide polymorphisms (SNPs), respectively [34,35]. Each SNP was documented based on the quantity of risk alleles and weighted risk estimate, with the products summed to generate the final genetic risk score. Our analyses included 183,381 participants with a completed genetic risk score for CHD and stroke. The genetic risk score was segmented into quintiles and subsequently classified into two groups: low risk (quintiles 1 to 4) and high risk (quintile 5) [34,35].

### 2.5. Definition of Outcomes

The primary outcomes for this study were all-cause mortality and incident CVD. The composite incident of CVD was characterized by CVD death, CHD, and stroke. The secondary outcome was CVD death, CHD, or stroke, respectively.

All-cause mortality was determined using the date of death ascertained from national death registries, including the National Health Service Information Centre for England and Wales and the National Health Service Central Register Scotland. Data were collected up to 1 January 2021. We therefore censored all-cause mortality analysis at this date or at date of death, whichever occurred first. A CVD event was defined as the first occurrence of CVD death, CHD, or stroke.

Codes from the International classification of diseases, 9th and 10th revision (ICD-9 and ICD-10) were used to assess the occurrence of disease. CVD death was defined by ICD-10 codes I00-I99. CHD was defined by ICD-9 codes 410–414 and ICD-10 codes I20–I25. Stroke was defined by ICD-9 codes 430–434, 436 and ICD-10 codes I60–I64, I678, I690 and I693.

### 2.6. Statistical Analysis

The baseline characteristics of the study sample were summarized by quintiles of dietary polyamines. Categorical variables are expressed as percentages, while normally distributed continuous variables are reported as mean and standard deviation. An analysis of variance or χ^2^ test was used to assess the difference between groups across the quintile, depending on the situation.

The relationship between dietary polyamines, SPD, SPM and PUT and all-cause mortality or incident CVD was estimated by Cox proportional hazard regression models. The results were presented by hazard ratios (HRs) and the corresponding 95% confidence intervals (CIs). Two models were employed: Model 1, adjusted for age and sex only; Model 2, adjusted for age, sex, ethnicity, education level, TDI, systolic blood pressure, BMI, physical activity, smoking status, alcohol status, sleep duration, energy, hypertension, diabetes, hypercholesteremia, antihypertensive treatment, lipid treatment and insulin treatment. The relationship between dietary polyamines, SPD, SPM and PUT and all-cause mortality or incident CVD was also assessed using restricted cubic spline (RCS) models. The RCS models were also conducted for the endpoint of CVD death, incident CHD and stroke, respectively. Knots were placed at the 25th, 50th, 75th and 100th percentiles of the exposure distribution. Missing data were handled by using a missing indicator for categorical covariates and by using median values for continuous covariates (all covariates <5% missing). Additional analyses were performed to assess the robustness of the results. Stratified analyses tested the potential correction effects by factors including age (<55 or ≥55), sex (female or male), TDI (least, moderate or most deprived), education (high level or other), BMI (normal, overweight or obese), smoking status (never, former or current), physical activity (low or high), sleep duration (>8 h/day, 7–8 h/day or <7 h/day), hypertension (no or yes), diabetes (no or yes), hypercholesteremia (no or yes), CHD genetic predisposition score (low or high) and stroke genetic predisposition score (low or high). Cox regression models were analyzed for multiplicative interactions by including a multiplicative term. Several sensitivity analyses were conducted as follows: (1) excluding participants who experienced the primary outcome within two years of the study period; (2) excluding participants with the top 5% and bottom 5% of polyamine levels; (3) adjusting for potential effect mediators (CHD and stroke genetic predisposition score); (4) excluding participants with any missing values of covariates. *p* values were 2-sided with statistical significance set at less than 0.05. All analyses were performed using R version 4.2.2.

## 3. Results

### 3.1. Baseline Characteristics

The baseline characteristics of participants from UK Biobank were shown in Table 1. Among 187,432 participants from UK biobank, the average age was 55.9 ± 7.9 years and 105,924 participants (56.5%) were female. Those with lower polyamines intake tended to be younger, female, non-white people, less educated, from the most deprived areas based on TDI and had a lower prevalence of comorbidities. Lower physical activity, unhealthy levels of smoking, obesity and inadequate sleep duration were more prevalent among the participants of lower polyamines intake.

### 3.2. Association of Dietary Polyamines with All-Cause Mortality and Incident CVD

During a median follow-up of 11.5 years, 7348 (3.9%) participants died and 12,316 (6.5%) had incident CVD. The all-cause mortality rates (cases per 1000 person-years) were 3.61, 3.04, 3.36, 3.47 and 3.58 for the quintile 1 to 5 group of dietary polyamines intake, while for incident CVD, they were 5.91, 5.4, 5.65, 5.68 and 6.58, respectively (Figure 1). Figure 1 and Figure 2 demonstrate the fully adjusted associations between polyamines intake and all-cause mortality or incident CVD. The RCS model flexibly demonstrates the relationship between polyamines and endpoint events. Moderate increases in polyamines were associated with a substantial reduction in the risk of all-cause mortality and incident CVD (Figure 2, nonlinear *p* < 0.001). Similar results were found when polyamines were treated as categorical variables. Compared to the quintile 1 group of dietary polyamines intake (≤17.4 mg/day), the quintile 2 to 5 groups demonstrated a consistent risk reduction in all-cause mortality, with the quintile 2 group having the lowest risk (>17.4–22.3 mg/day) (HR:0.82, 95% CI: 0.76–0.88). This pattern was also seen for incident CVD, with the lowest risk reduction in the quintile 4 group (>27.1–33.5 mg/day) (HR: 0.86, 95% CI: 0.82–0.92). Optimal polyamine intake for CVD components was observed in quintile 2 (>17.4–22.3 mg/day) for CVD death and quintile 4 (>27.1–33.5 mg/day) for incident CHD and stroke.

We further analyzed the impact of polyamine composition on endpoints. The RCS models were used to analyze the significant nonlinear relationships between dietary SPD, SPM, PUT and all-cause mortality and incident CVD (Appendix A). The benefit of dietary SPD intake to all-cause mortality was most pronounced in the quintile 2 group (>6.1–7.9 mg/day) and then leveled off, a pattern also observed for incident CVD (Appendix A). The optimal benefit range for all-cause mortality was (quintile 3: >3.4–4.3 mg/day) for SPM and (quintile 2: >7.1–9.9 mg/day) for PUT, while for incident CVD, it was (quintile 3: >3.4–4.3 mg/day) for SPM and (quintile 3: >9.9–12.8 mg/day) for PUT (Appendix A).

Stratified analyses were performed based on potential risk factors for CVD. The inverse relationship between polyamines intake and endpoints remained consistent across sex, age and various subgroups. Significant interactions were found between polyamines intake and education level, sleep duration and hypercholesteremia on all-cause mortality (Interaction *p* = 0.027, 0.047 and 0.025, respectively) (Figure 3). The associations between dietary polyamines intake and endpoints were stronger among those with higher education level, shorter sleep duration or without hypercholesteremia. For incident CVD, the associations were stronger among those with higher education level, without diabetes, or with lower CHD and stroke genetic predisposition scores. (Interaction *p* = 0.031, 0.015 and <0.001, respectively) (Figure 3). We also found significant interactions between polyamines intake and BMI in CVD death, as well as education level, smoking status and CHD genetic predisposition score in incident CHD and sleep duration in incident stroke (Appendix A).

Sensitivity analyses showed that the associations between dietary polyamines and endpoints remained stable after several adjustments: first, excluding participants who experienced the primary outcome within two years of the study period (Appendix A); second, excluding participants with the bottom 5% and top 5% of polyamine levels (Appendix A); third, adjusting for potential effect mediators (CHD and stroke genetic predisposition score) (Appendix A) and fourth, excluding participants with any missing covariate value (Appendix A).

## 4. Discussion

To our knowledge, our study is the first to demonstrate an inverse correlation between dietary polyamines intake and all-cause mortality and incident CVD in the general population. Based on the prospective study in UK Biobank, moderate polyamines intake was associated with an 18% lower risk of all-cause mortality and a 14% reduction in total CVD events (CVD death, CHD and stroke). The correlation was found to be independent of traditional risk factors affecting lifespan and habits, such as sex, age, ethnicity, TDI, education, BMI, physical activity, smoking status, alcohol intake, energy, systolic blood pressure, diabetes, hypertension, hypercholesteremia and drug use (antihypertensive, lipid and insulin treatment). Furthermore, we found that the associations between polyamines and primary endpoints were statistically significant with the effect of education level. These findings may further complement the epidemiological evidence about the effects of polyamines level on mortality and incident CVD, while their correlations with optimal dosage offer insights into dietary guidelines or replacement therapy.

Our findings are partly consistent with prior studies. In the Bruneck study, involving 829 Italian participants, showed that as SPD intake increased, all-cause mortality (deaths per 1000 person-years) decreased from 40.5 (95% CI 36.1, 44.7) to 23.7 (95% CI 20.0, 27.0) and 15.1 (95% CI 12.6, 17.8); a similar result was found for SPM intake, whereas PUT did not show a correlation with endpoints [13]. Another study further validated the role of SPD in all-cause mortality and further explored its effect on CVD mortality using data from the NHANES study, showing improved prognosis with dietary SPD from various sources [14]. However, in the Takayama study of 29,079 participants, the results showed no significant correlation between polyamine intake and all-cause or cause-specific mortality [15]. Our study revealed that polyamines intake was associated with a lower risk of all-cause mortality and incident CVD, with the greatest observed in the moderate range. The nonlinear relationship between polyamine intake and endpoints found in our study differs somewhat from several previous studies, likely due to variations in population structure and methods of dietary polyamines assessment. The retrospective recall of dietary components can introduce misclassification, as dietary habits may change over time due to factors such as age, job or economic status. To minimize residual confounding related to lifestyle and dietary patterns associated with polyamine intake, our study uniquely included data from up to 240 dietary types combined with up to five occurrences of dietary assessments to obtain more accurate data on dietary polyamines; it is also the largest population-based study of polyamines that is different from the Bruneck study. We also found that in the NHANES study, the average SPD intake was 350 umol/day, much higher than in other current studies (Bruneck study: average SPD intake: 71 umol/day; Takayama study: average SPD intake: 95.7 umol/day; our study: average SPD intake: 64.4 umol/day), which might explain why our results differ from the NHANES study [13,14,15].

Several potential pathophysiologic mechanisms may explain the association between polyamines and mortality or CVD. Animals studies have shown that SPD, a natural polyamine, can extend the lifespan in various organisms by promoting autophagy [8,10]. Recent studies indicate that autophagy plays a crucial role in maintaining the quality of aging cardiomyocytes by recycling long-lived proteins and damaged organelles [36,37]. Notably, autophagy is required for SPD-mediated cardioprotection and SPD enhances cardiomyocyte autophagic flux. These beneficial effects were found in both young and aged mice, indicating that although autophagy and SPD showed a significant age-related decrease, younger individuals can also obtain benefits from SPD supplementation [11,38]. Our study was similar in the population, where polyamines were stable at the endpoint when stratified by age. A previous study indicated that the oral supplementation of SPD was linked to reduced salt-induced hypertension, anti-inflammatory effects and the mitigation of mechano-elastical impairments and mitochondrial dysfunction associated with hypertension and chronic low-grade inflammation [39]. Our study revealed the optimal doses of SPD, SPM and PUT for beneficial effects, but the results of PUT were different from the previous studies. When compared with the mice supplemented with SPD or SPM, PUT supplementation or control could not extend the median lifespan [10]. Similar results were found in the Bruneck study [13]. Although 60–80% of the infused PUT disappeared from the lumen linearly with time, its metabolically converted succinate can act as an energy source for the gut [40]. It is important to note that in our daily diet including a mixture of amines, rather than a specific amine, PUT differs from SPD in its genesis and metabolic reactions; our results motivated our interest in whether appropriate oral PUT has other benefits beyond improving intestinal energy supply.

Our study has several strengths. Firstly, since it is based on a prospective cohort featuring a large sample size and long-term follow-up, there are a considerable number of cases for our study to assess the association between polyamines and primary endpoints. Secondly, repeated dietary questionnaire assessment and the comprehensive information on lifestyle, energy, medical history and other covariates enable us to provide a rationalized range of dietary polyamines intake. Thirdly, in addition to polyamines, we fully evaluated the effect of the composition of polyamines on the clinical outcomes. Finally, sensitivity and subgroup analyses, along with the addition of polygenic risk scores, further refine our results. However, we acknowledge limitations in our study. First, dietary assessments relied on 24 h recall, which may not accurately reflect participants’ usual intake, especially considering the presence of extreme values. Although we have excluded such data, there may be recallable bias. Second, our study is also subject to limitations inherent in nutritional epidemiology, such as the nonconsideration of storage conditions and food preparation for each food, which would be beyond the scope of WebQ but may affect polyamines content. It is also important to note that the question of whether oral polyamine supplementation increases blood polyamines levels remains debated. SPD-fed animals showed elevated levels of SPD in their blood stream, while healthy Japanese men experienced an increase in blood SPD levels after consuming a diet rich in SPD from fermented soybeans over a prolonged period [11,41]. However, there was no connection between dietary polyamines and urinary polyamines levels [42], warranting further investigation. Third, despite controlling for numerous widely accepted confounders, we cannot avoid the possibility of residual or unmeasured confounding effects. For instance, although we adjusted for energy, BMI related to state of nutrition in our analysis model and the effects of other nutrients were not discussed in this study. Fourth, this nonlinear correlation disclosed by regression modeling may require continued support from substantial evidence. Finally, the majority of UK Biobank participants are of Caucasian descent, thereby limiting the generalizability of our findings to diverse geographic regions or ethnic groups.

## 5. Conclusions

In summary, dietary polyamines intake was associated with a lower risk of all-cause mortality and incident CVD. Similar results were found in SPD, SPM and PUT. Further analysis indicates that the associations between dietary polyamines and endpoints remained stable after adjusting for CHD and stroke genetic predisposition scores. This study offers population-based evidence to substantiate the value of dietary polyamines and also identified an optimal range of dietary polyamines intake, which may inform future dietary recommendations and alternative oral polyamines supplements.

## Figures and Tables

**Figure 1 nutrients-16-04335-f001:**
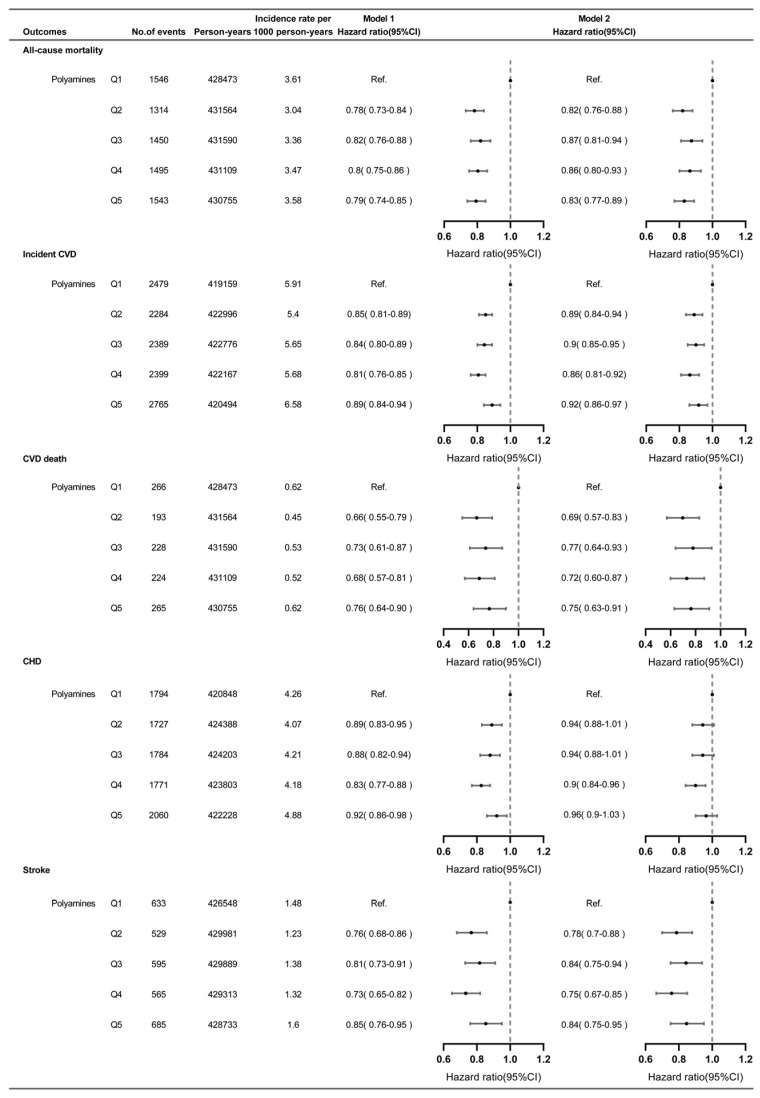
The association between dietary polyamines intake and all-cause mortality or incident CVD and components of CVD (CVD death, CHD and stroke). Mode1 1: Analyses adjusted for age and sex. Model 2: Analyses adjusted for age, sex, ethnicity, Townsend deprivation index, education level, systolic blood pressure, body mass index, physical activity, smoking status, alcohol status, sleep duration, energy, hypertension, diabetes, hypercholesteremia, antihypertensive treatment, lipid treatment and insulin treatment. Indications: quintile 2 to 5 groups demonstrated a consistent risk reduction in all-cause mortality, with the quintile 2 group having the lowest risk (>17.4–22.3 mg/day). This pattern was also seen for incident CVD, with the lowest risk reduction in quintile 4 group (>27.1–33.5 mg/day). Optimal polyamine intake for CVD components was observed in quintile 2 (>17.4–22.3 mg/day) for CVD death and quintile 4 (>27.1–33.5 mg/day) for incident CHD and stroke.

**Figure 2 nutrients-16-04335-f002:**
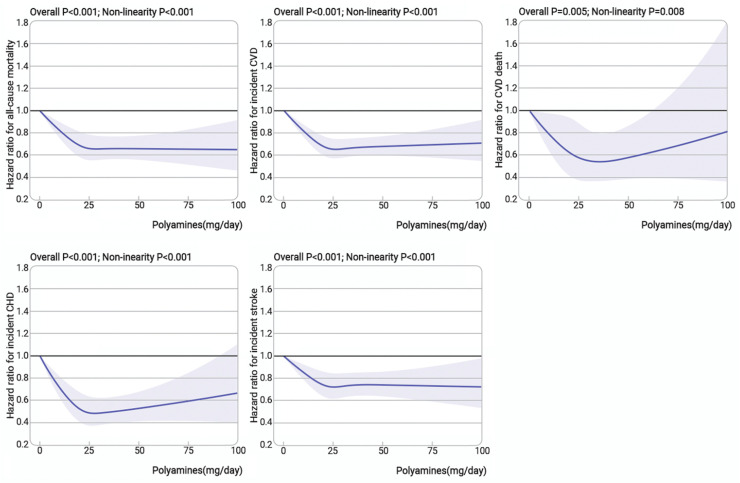
The association between dietary polyamines intake and all-cause mortality or incident CVD. Knots were placed at the 25th, 50th, 75th and 100th percentiles of the polyamines intake distribution. Analyses adjusted for age, sex, ethnicity, Townsend deprivation index, education level, systolic blood pressure, body mass index, physical activity, smoking status, alcohol status, sleep duration, energy, hypertension, diabetes, hypercholesteremia, antihypertensive treatment, lipid treatment and insulin treatment. Components of CVD (CVD death, CHD and stroke) were also analyzed. Shaded areas represent 95% confidence intervals. Indications: Moderate increases in polyamines were associated with a substantial reduction in the risk of all-cause mortality and incident CVD. (Nonlinear *p* < 0.001). Similar results were found in the composition of CVD events.

**Figure 3 nutrients-16-04335-f003:**
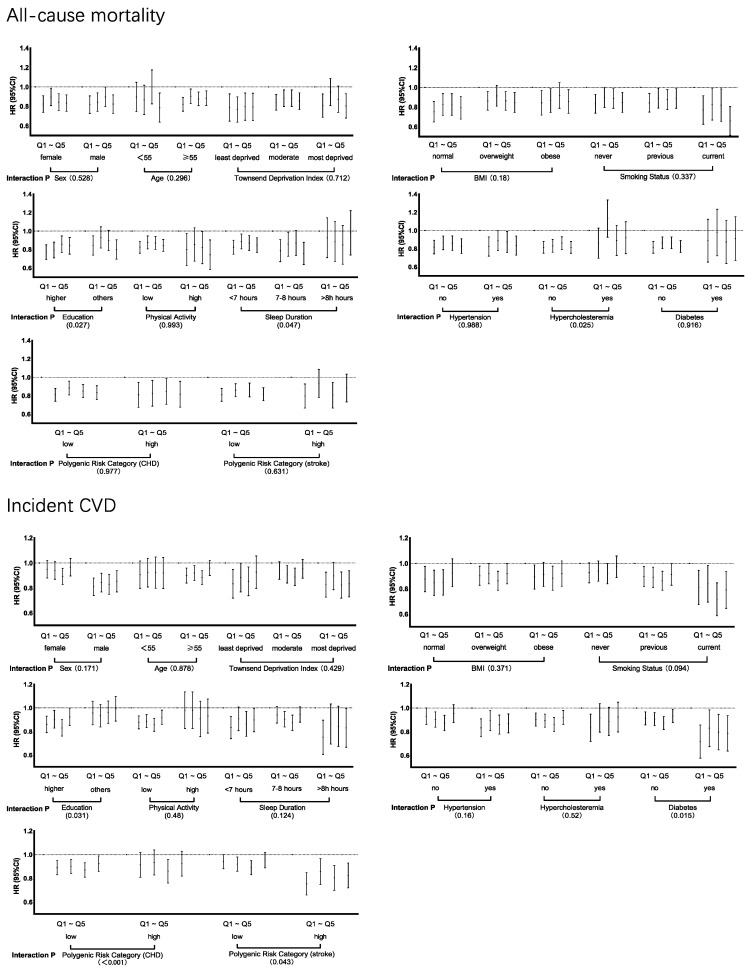
Association between dietary polyamines intake and all-cause mortality or incident CVD stratified by potential risk factors. Analyses adjusted for age, sex, ethnicity, Townsend deprivation index, education level, systolic blood pressure, body mass index, physical activity, smoking status, alcohol status, sleep duration, energy, hypertension, diabetes, hypercholesteremia, antihypertensive treatment, lipid treatment and insulin treatment. Indications: Significant interactions were found between polyamines intake and education level, sleep duration and hypercholesteremia on all-cause mortality; for incident CVD, the associations were stronger among those with a higher education level, without diabetes, or with lower CHD and stroke genetic predisposition scores.

**Table 1 nutrients-16-04335-t001:** Baseline characteristics of UK Biobank participants by dietary polyamines quintile.

Characteristics	Total n = 187,432	Quintile Groups of Polyamines (mg/d)					*p* Value
		Quintile 1 (≤17.4) n = 37,487	Quintile 2 (>17.4–22.3) n = 37,486	Quintile 3 (>22.3–27.1) n = 37,487	Quintile 4 (>27.1–33.5) n = 37,486	Quintile 5 (>33.5–98.9) n = 37,487	
Age, mean (SD), y	55.9 (7.92)	54.64 (8.03)	55.43 (7.96)	56.04 (7.87)	56.57 (7.78)	56.93 (7.77)	<0.001
Sex, female, n (%)	105,924 (56.5)	22,373 (59.7)	21,872 (58.3)	21,419 (57.1)	20,763 (55.4)	19,497 (52.0)	<0.001
Ethnicity, n (%)							<0.001
White European	179,239 (95.6)	35,132 (93.7)	35,941 (95.9)	36,148 (96.4)	36,178 (96.5)	35,840 (95.6)	
Mixed	1120 (0.6)	299 (0.8)	207 (0.6)	219 (0.6)	194 (0.5)	201 (0.5)	
Asian	2958 (1.6)	871 (2.3)	580 (1.5)	488 (1.3)	460 (1.2)	559 (1.5)	
Black	2155 (1.1)	697 (1.9)	376 (1.0)	305 (0.8)	326 (0.9)	451 (1.2)	
Others	1960 (1.0)	488 (1.3)	382 (1.0)	326 (0.9)	328 (0.9)	436 (1.2)	
Townsend deprivation index ^a^							<0.001
Least deprived	37,547 (20.0)	6690 (17.8)	7350 (19.6)	7943 (21.2)	7950 (21.2)	7614 (20.3)	
Moderate deprived	112,399 (60.0)	21,769 (58.1)	22,542 (60.1)	22,549 (60.1)	22,756 (60.7)	22,783 (60.8)	
Most deprived	37,486 (20.0)	9028 (24.1)	7594 (20.3)	6994 (18.7)	6780 (18.1)	7090 (18.9)	
Education, n (%)							<0.001
College or university degree	80,846 (43.1)	14,156 (37.8)	16,299 (43.5)	16,830 (44.9)	17,099 (45.6)	16,462 (43.9)	
Others	90,537 (48.3)	19,243 (51.3)	18,162 (48.4)	17,783 (47.4)	17,626 (47.0)	17,723 (47.3)	
Unknown	16,049 (8.6)	4088 (10.9)	3025 (8.1)	2873 (7.7)	2761 (7.4)	3302 (8.8)	
Systolic Blood Pressure, mmHg		135.08 (18.11)	135.95 (18.23)	136.61 (18.20)	137.37 (18.24)	138.11 (18.31)	<0.001
Body mass index, n (%)							<0.001
Underweight (<18.5)	954 (0.5)	187 (0.5)	183 (0.5)	197 (0.5)	201 (0.5)	186 (0.5)	
Normal weight (18.5 to <25)	68,801 (36.7)	13,199 (35.2)	14,041 (37.5)	14,134 (37.7)	14,122 (37.7)	13,305 (35.5)	
Overweight (25 to <30)	79,994 (42.7)	15,967 (42.6)	16,096 (42.9)	15,947 (42.6)	15,945 (42.5)	16,039 (42.8)	
Obese ≥ 30	37,683 (20.1)	8134 (21.7)	7166 (19.1)	7208 (19.2)	7218 (19.3)	7957 (21.2)	
Physical activity, n (%) ^b^							<0.001
Low	164,138 (87.6)	32,967 (87.9)	33,321 (88.9)	33,072 (88.2)	32,878 (87.7)	31,900 (85.1)	
High	23,294 (12.4)	4520 (12.1)	4165 (11.1)	4414 (11.8)	4608 (12.3)	5587 (14.9)	
Smoking, n (%)							<0.001
Never	107,387 (57.3)	21,057 (56.2)	21,488 (57.3)	21,791 (58.1)	21,846 (58.3)	21,205 (56.6)	
Former	65,177 (34.8)	12,264 (32.7)	12,963 (34.6)	12,979 (34.6)	13,255 (35.4)	13,716 (36.6)	
Current	14,393 (7.7)	4048 (10.8)	2950 (7.9)	2628 (7.1)	2295 (6.1)	2472 (6.6)	
Unknown	475 (0.2)	118 (0.3)	85 (0.2)	88 (0.2)	90 (0.2)	94 (0.2)	
Alcohol, n (%)							<0.001
Never	5840 (3.1)	1423 (3.8)	1120 (3.0)	1087 (2.9)	1017 (2.7)	1193 (3.2)	
Former	5323 (2.8)	1349 (3.6)	1058 (2.8)	886 (2.4)	965 (2.6)	1065 (2.8)	
Current	176,096 (94.0)	34,667 (92.5)	35,265 (94.1)	35,487 (94.6)	35,478 (94.6)	35,199 (93.9)	
Unknown	173 (0.1)	48 (0.1)	43 (0.1)	26 (0.1)	26 (0.1)	30 (0.1)	
Sleep duration, n (%)							<0.001
<7 h	41,887 (22.3)	9097 (24.3)	8218 (21.9)	8005 (21.4)	8005 (21.4)	8562 (22.8)	
7–8 h	133,303 (71.1)	25,686 (68.5)	26,774 (71.4)	27,180 (72.5)	27,113 (72.3)	26,550 (70.8)	
>8 h	11,670 (6.3)	2528 (6.7)	2388 (6.4)	2215 (5.9)	2265 (6.0)	2274 (6.1)	
Unknown	572 (0.3)	176 (0.5)	106 (0.3)	86 (0.2)	103 (0.3)	101 (0.3)	
Polyamines (SD), mg/day							
Spermidine	9.35 (4.78)	5.05 (1.72)	7.45 (1.62)	8.98 (1.85)	10.54 (2.23)	14.71 (6.91)	<0.001
Spermine	4.18 (1.93)	2.59 (1.16)	3.58 (1.31)	4.18 (1.50)	4.78 (1.74)	5.75 (2.14)	<0.001
Putrescine	12.3 (6.36)	5.44 (2.24)	8.90 (2.22)	11.49 (2.59)	14.68 (3.23)	21.10 (5.72)	<0.001
Energy, mean (SD), KJ	8590 (2240)	7311.85 (2024.45)	8194.87 (1948.86)	8660.61 (1990.94)	9060.59 (2089.97)	9742.72 (2326.27)	<0.001
Hypertension, n (%)	41,894 (22.4)	7965 (21.2)	8078 (21.5)	8139 (21.7)	8459 (22.6)	9253 (24.7)	<0.001
Diabetes, n (%)	6593 (3.5)	1233 (3.3)	1218 (3.2)	1192 (3.2)	1311 (3.5)	1639 (4.4)	<0.001
Hypercholesteremia, n (%)	18,787 (10.0)	3616 (9.6)	3605 (9.6)	3586 (9.6)	3842 (10.2)	4138 (11.0)	<0.001
Drug use, n (%)							
Antihypertensive	30,384 (16.2)	5583 (14.9)	5753 (15.3)	5979 (15.9)	6264 (16.7)	6805 (18.2)	<0.001
Lipid treatment	24,005 (12.8)	4526 (12.1)	4573 (12.2)	4618 (12.3)	4941 (13.2)	5347 (14.3)	<0.001
Insulin treatment	1342 (0.7)	246 (0.7)	254 (0.7)	227 (0.6)	266 (0.7)	349 (0.9)	<0.001
Genetic predisposition score ^†^							
CHD predisposition score	12.2 (0.483)	12.2 (0.484)	12.2 (0.480)	12.2 (0.483)	12.2 (0.485)	12.2 (0.482)	0.11
Stroke predisposition score	1.83 (0.359)	1.83 (0.361)	1.83 (0.359)	1.82 (0.358)	1.82 (0.357)	1.83 (0.359)	0.074

CHD: coronary heart disease. ^a^: Townsend deprivation index: least: 1st quintile, moderate: 2–4, most: 5th quintile. ^b^: Physical activity: low: <1200 (Metabolic Equivalent Task (MET) minutes per week for all activity) and high (≥1200 MET-min/week). ^†^ Data were available for 183,381 participants.

## Data Availability

Researchers can apply to use the UK Biobank resource and access the data used. The UK Biobank data are available on application to the UK Biobank (https://www.ukbiobank.ac.uk/, accessed on 1 January 2021).

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
