# Peer review of "The Association of Dietary Polyamines with Mortality and the Risk of Cardiovascular Disease: A Prospective Study in UK Biobank"

_nutrients, 2024, doi:10.3390/nu16244335_

Round 1
Reviewer 1 Report
Comments and Suggestions for Authors
Journal Nutrients (ISSN 2072-6643)
Manuscript ID nutrients-3318109
Type Article
Title Association of dietary polyamines with mortality and the risk of cardiovascular disease: prospective study in UK Biobank
Authors Su Han , Mingxia Qian , Na Zhang , Rui Zhang , Min Liu , Jiangbo Wang , Furong Li , Liqiang Zheng * , Zhaoqing Sun *
Section Nutrition and Public Health
Special Issue Diet, Nutrition and Cardiovascular Health
Dear Editor
The text of the manuscript requires thorough proofreading in terms of content and editing.
I enclose my comments below.
With best regards.
Reviewer
KEY WORDS
1.
Please eliminate terms appearing in the title of the manuscript.
INTRODUCTION
2.
l. 37 - 67. Authors should enter the correct citation according to the Guidelines for Authors. The record of publication citation indicates a lack of information about this editorial action.
l. 51 “7-9 Eisenberg T et al.”
l. 58 “ Kiechl S et al.”
l. 59, 61, 62.
3.
l. 38 - 43. broad concepts, please provide details instead of generalities known for years.
- Please complete the factual information “Cardiovascular disease”.
- State what and as a result of what - metabolic and biochemical processes “cause of mortality and disability world wide.”
4.
l. 40-43. What the authors want to convey please provide detailed and factual information. What ingredients, caloric value, biologically active compounds, ratio of fatty acids from the group: omega e.g. omega 3, omega 6, omega and specific chemical compounds from antioxidants, vitamins, polyphenolic compounds including anthocyanins etc.) . The concept of diet is broad. Please rewrite the text with reference to the manuscript topic.
5.
l. 44 - 47. Please describe these metabolic processes, taking into account a number of specific proteins referred to as transcription factors, responsible for the induction of the transcription process in close reference to the topic of the manuscript.
6.
l. 47-51. Describe the biosynthesis of polyamines, among others.
- in plants, the essential source of putrescine is the urea cycle
- decarboxylation of arginine to agmatine and further via N-carbamoyl-putrescine to putrescine
- citrulline, after decarboxylation and detachment of the -NH2 group as NH3, is converted to putrescine
7.
l. 47-51. instead of the enumeration ‘DNA stability, transcription, translation, and apoptosis.’ These are complex and important metabolic processes. Please explain substantively these running processes in the body with reference to the topic of the manuscript.
8.
51-60.
Please provide for each example the biological model of the experiment, dose, mode of application, population size, age and main conclusion in close reference to the manuscript topic
9.
l. 61-63. Please state these limitations.
10.
l. 64-67. please specifically state what has been done in this area and where there is a gap in the literature.
11.
l. 64-67. some sentence fragments not in this place.
12.
Please justify the need for the research presented.
13.
Please formulate the specific aim of the work.
METHODS
14.
Methods should be described in such detail that everything can be understood with the ability to open the procedure without looking at other sources. Please make a correction.
RESULTS
15.
Figure 1: Please indicate the difference between the diagrams in the description. Markings are illegible (too low magnification)
16.
Figure 2. Please indicate the resulting specific information in the description. Indications are illegible (too low magnification).
17.
Fugure 3. See comments 15 and 16.
DISCUSION
18.
Please use correct publication citation in your discussion.
19.
Reduce the nature of the review to discussion.
20.
Complete the applicability of the research results obtained.
21.
Indicate the novelty of the research results obtained.
22.
Complete the prospect of future research.
23
l. 252-231. authors should introduce correct citation according to the guidelines for authors.
CONCLUSIONS
24.
Please state concretely what conclusions are drawn from the study, from each methodological step.
25.
Conclusions should respond to the aim of the paper and the scientific research theses set. Please rephrase.
REFERENCES
26.
There is an inconsistency with the Guidelines for Authors in each literature item. Please check the assignment sentence by sentence and make a change in accordance with the guidelines for authors when compiling the literature list, e.g. item 1 - 21, further on I do not check. It is impossible to list everything.
Reviewer 2 Report
Comments and Suggestions for Authors
The paper is well-written. The methods meet today's standards for this type of study. The inclusion and exclusion criteria were adequate and sufficiently described. The data presentation is well articulated and easily followed.
Author Response
Thank you for recognizing our research.
Reviewer 3 Report
Comments and Suggestions for Authors
1) Authors described that "To address these research debates and gaps, we examined the relationship between dietary polyamines (SPD, SPM and PUT) and all-cause mortality or incident CVD in a population of approximately 500,000 participants in the UK Biobank. Additionally, we examined the potential influence of other established risk factors for CVD on the results. " However, I could not understand this manuscript's hypothesis, so could you add this more clearly?
2) Selection of the subjects we not clear, it needs to be revised to add the flow chart.
3) Too short of the background in the present this study, why does authors try to do this study? Could you show that more clearly?
4) Is this statistical method appropriate? Please consider again.
5) What process was used to select the subjects for this final analysis? It's not clear. Could this selection be reflected in the population?
6) In the Table1, there were only values, so could you add the more information here?
7) Too small of the tables and figures, so not easy to understand for readers, so could you revise this more clearly?
8) Is this consideration relevant? Consider based on the results of this study as much as possible. In addition, I think that there is a lot of presentation of the results and little consideration.
9)I'm sure there are many other limitations of the study, but please consider it again.
10)Is the citation method of the literature reasonable? Again, please consider including the submission rules.
Comments on the Quality of English Languageit is better to have the English proofread again.
Round 2
Reviewer 3 Report
Comments and Suggestions for Authors
There are no more comments for this.